# Study on the enhancement of low carbon-to-nitrogen ratio urban wastewater pollutant removal efficiency by adding sulfur electron acceptors

Erming Luo[1], Jia Ouyang[1], Xinxin Zhang[1], Qian Lu[1], Dong Wei[2,3], Yongcheng Wang[4], Zhengjiong Cha[4], Chengwei Ye[4], Chun ying Li[5]*, Li Wei[1,2]*

1 Guangzhou HKUST Fok Ying Tung Research Institute, Guang Zhou, China, 2 State Key Laboratory of Urban Water Resource and Environment, Harbin Institute of Technology, Harbin, Heilongjiang Province, China, 3 College of Life Sciences, Northeast Forestry University, Harbin, Heilongjiang, China, 4 Guangzhou COSMO Environment Technology CO.,LTD, Guang Zhou, China, 5 School of Energy and Civil Engineering, Harbin University of Commerce, Harbin, China

* heart.li@163.com (CYL); weilihkust@ust.hk (LW)

**Data Availability Statement:** All relevant data are within the paper.

**Funding:** This study was funded by the following grants: Funder Name: the Guangdong Provincial

## Abstract

The effective elimination of nitrogen and phosphorus in urban sewage treatment was always hindered by the deficiency of organic carbon in the low C/N ratio wastewater. To overcome this organic-dependent barrier and investigate community changes after sulfur electron addition. In this study, we conducted a simulated urban wastewater treatment plant (WWTP) bioreactor by using sodium sulfate as an electron acceptor to explore the removal efficiency of characteristic pollutants before and after the addition of sulfur electron acceptor. In the actual operation of 90 days, the removal rate of sulfur electrons' chemical oxygen demand (COD), ammonia nitrogen, and total phosphorus (TP) with sulfur electrons increased to 94.0%, 92.1% and 74%, respectively, compared with before the addition of sulfur electron acceptor. Compared with no added sulfur(phase I), the reactor after adding sulfur electron acceptor(phase II) was demonstrated more robust in nitrogen removal in the case of low C/N influent. the effluent ammonia nitrogen concentration of the aerobic reactor in Pahse II was kept lower than 1.844 mg N / L after day 40 and the overall concentration of total phosphorus in phase II (0.35 mg P/L) was lower than that of phase I(0.76 mg P/L). The microbial community analysis indicates that *Rhodanobacter*, *Bacteroidetes*, and *Thiobacillus*, which were the predominant bacteria in the reactor, may play a crucial role in inorganic nitrogen removal, complex organic degradation, and autotrophic denitrification under the stress of low carbon and nitrogen ratios. This leads to the formation of a distinctive microbial community structure influenced by the sulfur electron receptor and its composition. This study contributes to further development of urban low-carbon-nitrogen ratio wastewater efficient and low-cost wastewater treatment technology.

Science and Technology Planning Project Grant No.: 2023A0505030018 Funding Recipient: Dr. Li Wei Funder Name: Guangzhou Municipal Science and Technology Project Grant No.: 202201011743 Funding Recipient: Erming Luo Funder Name: Guangzhou Municipal Science and Technology Project Grant No.: 202201011683 Funding Recipient: Jia Ouyang Funder Name: Guangzhou Municipal Science and Technology Project Grant No.: 202201011584 Funding Recipient: Xinxin Zhang Funder Name: Guangzhou Municipal Science and Technology Project Grant No.: 2024A03J0392 Funding Recipient: Jia Ouyang Funder Name: the International Science and Technology Cooperation Project of Huangpu District, Guangzhou Grant No.: 2020GH04 Funding Recipient: Dr. Li Wei Funder Name: the Science and Technology project of Nansha District, Guangzhou Grant No.: 2023ZD015 Funding Recipient: Dr. Li Wei Funder Name: the Science and Technology project of Nansha District, Guangzhou Grant No.: 2023ZD021 Funding Recipient: Dr. Li Wei Funder Name: the Science and Technology project of Nansha District, Guangzhou Grant No.: 2021MS017 Funding Recipient: Dr. Li Wei.

**Competing interests:** The authors have declared that no competing interests exist.

## 1. Introduction

Nowadays, the excessive discharge of nitrogen and phosphorus [1] in wastewater elevated the treatment challenges faced by urban wastewater treatment plants (WWTPs).such as eutrophication [2] and low C/N ratio wastewater. Carbon to nitrogen ratio (C/N) was defined as the ratio of chemical oxygen demand (COD) to nitrogen. The C/N ratios of polluted surface water, groundwater, and wastewater treatment plant tailwater are generally below 5 [3],which increased the difficult of biological nitrogen removal. Although Biological Nitrogen Removal (BNR) systems in WWTPs played a key role in reducing nitrogen and phosphorus discharges, the effective removal of nitrogen and phosphorus in the BNR process was still limited by organic carbon deficiencies [4]. While various external carbon sources, such as methanol, sodium acetate, or synthetic carbon sources, had been widely utilized, the challenge lies in the accurate control of their release speed and dosage. This difficulty not only had the potential to impact the effluent quality of sewage treatment plants and increased the cost of sewage treatment but can also pose safety concerns during transportation and storage [5]. Hence, within urban WWTPs, addressing the challenge of achieving nitrogen and phosphorus removal without augmenting external carbon addition and increasing energy consumption remains a noteworthy hurdle for low C/N ratio wastewater [6].

During the process of BNR nitrification and denitrification are the primary nitrogen removal methods [7]. Biological nitrogen removal can be categorized into two types: heterotrophic microbe-driven processes, utilizing sufficiency organic waste and external carbon sources for denitrification, while autotrophic microbe-driven processes, which using inorganic carbon and compounds like hydrogen, ferrous ions, and sulfur as electron donors for denitrification and anaerobic ammonia oxidation [8, 9].

Therefore, autotrophic microbe-driven processes had been proposed in to wastewater treatment such as Single Reactor with Highly Active Ammonia for Nitrite Removal (SHARON) and Anaerobic Ammonia Oxidation (Anammox) [10, 11]. SHARON could oxidised ammonium into nitrite instead of nitrate and Anammox uses nitrite as an electron acceptor for nitrogen to convert ammonium. Each of these processes saves up to 80% in operating costs [12, 13]. This is due to the absence of organic carbon and reduced sludge production [14]. However, due to the stringent requirements for temperature, pH, alkalinity, dissolved oxygen, it was still a challenge to obtained and cultured the sufficient and stable ammonium-oxidising bacteria (AOB) in the SHARON (e.g., *Nitrosomonas europaea*, *Nitrosomonas eutropha*) and anaerobic AOBs in Anammox(e.g., *Candidatus Brocadia anammoxidans* and *Candidatus Kuenenia stuttgartiensis*) [15, 16].

Because sulfur has multiple valence, it was a good oxidant or reducing agent, while the conversion of sulfur in the environment seriously depends on the activities of microorganisms. Microbial transformation of inorganic sulfide and organic sulfur compounds has had a profound impact on the properties of the biosphere [17]. Based on the phenomenon of low organic carbon source in urban wastewater, the use of sulfur autotrophic microorganism-driven autotrophic denitrification to remove nutrients has become one of the most attractive schemes for treating low C / N wastewater. It avoided increased biosludge production, $NO_2$ accumulation and emissions for over-addition of carbon sources [18]. Sodium sulphate, a readily available, less toxic and inexpensive form of sulphur [19], could be used as an electron acceptor for organic matter removal, and the sulphide produced can further be used as an electron donor for autotrophic denitrification to remove nitrate from water [20].

In order to overcome the disadvantages of urban low C/N ratio, the low C/N ratio nitrogen removal process in wastewater based on sulfur as an electron acceptor was developed. The sludge in the secondary clarifier was directly recycled to the anaerobic selector, and the denitrification liquid was recycled from anoxic to anaerobic. To extend the sludge retention time

 

(SRT) of nitrifying bacteria, square fillers were added in the aerobic bioreactor. The design aims to improve anaerobic phosphorus release, Denitrifying Phosphate Accumulating Organisms(DPAO) enrichment, and inhibit filamentous sludge bulking [21]. It can be easily constructed and adapted from existing Anaerobic/Oixc (A/O) or Anaerobic/Anoxic/Oixc(AAO) processes by changing piping, aeration, HRT and microbial communities in BNR. This meets WWTPs requirements, facilitating the upgrade of existing A/O or AAO processes to enhance sewage treatment quality with low reconstruction costs. By cultivating and utilizing sulfur autotrophic denitrifying strains, urban sewage treatment plants can not only improve nitrogen removal efficiency in low carbon and nitrogen ratio wastewater [22], but also significantly reduce sludge production [23].

Based on the laboratory-scale study of the low-carbon-nitrogen removal process, we optimized the operating parameters of the process and developed good operating performance. We aimed to study: (i) the initiation of the sulfur electron acceptor process; (ii) the removal performance of various contaminants (COD, TP, $NH_4^+$-N and total nitrogen (TN)) and (iii) the change of microbial community structure in each functional area. This study provides a new idea for the removal of wastewater from urban sewage treatment plants.

## 2. Materials and methods

### 2.1 Reactor configuration and operating conditions

The experimental reactor was made of acrylic glass with an overall effective volume of 30L (Fig 1) and was operated at 25°C. The continuous flow low C/N ratio wastewater treatment process mainly consists of wastewater storage tank, sulfur electron receptor adding tank, secondary sedimentation tank, two Upflow Anaerobic Sludge Bed (UASB) reactors and an aerobic reactor. The two UASB reactor was divided into an anaerobic reactor and a anoxic reactor. The top of the reactor was sealed with a top cover with exhaust holes. The volume of anaerobic and anoxic reactor is 7 L each, while the aerobic reaction zone is 16 L.

The aeration speed was 10 L / h (DO concentration: 2.5–3 mg/L) in the middle of the aerobic reactor. Meanwhile, the aerobic reactor also used polyurethane filler (2 x 2 x 2cm) to replace the transformation of activated sludge method. This was due to its large specific surface area and good membrane effect, which could significantly improve the efficiency and stability of the biochemical system. The aerobic reactor returned part of the sewage into the anoxic reaction pool, the reflux ratio between the aerobic reactor and the anoxic reactor was 1:1, and the reflux ratio of sludge form the secondary sedimentation tank to anaerobic reactor was 50%. In this experiment, the Hydraulic Residence Time (HRT) of the anaerobic, anoxic, and aerobic reactor were set to 8h, 4h and 8h, respectively.

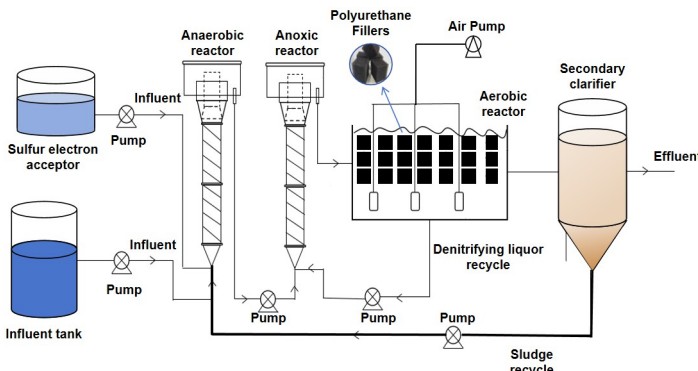

**Fig 1. The sulfur electron acceptor injection process.**

The reactor operation was divided into 2 phases: no added sulfur (phase I) and added sulfur electrons (phase II). To investigate the effect of sulfur electron acceptor on biological nitrogen removal by adding sulfur electron acceptor, the reactor was operated under normal stable conditions in phase I(days 1–30)and the 100 mg S/L of the sulfur electron acceptor was added into the synthetic sewage in phase II(days 31–90). The experimental parameters of phases I and II are shown below.

## 2.2 Inoculation sludge and synthetic sewage

In this study, the inoculated sludge came from a sewage treatment plant in Nansha, Guangzhou, which operated stably for a long time. Before the sludge was inoculated, the sludge was cleaned to remove impurities in the sludge. 5L of anaerobic sludge was inoculated in each of the two UASB bioreactors with a sludge concentration of about 4000 to 5500 mg / L. At the same time, 3L aerobic sludge was inoculated in the aerobic reactor and 30% polyurethane fillers was added to the aerobic sludge for mixing.

In order to investigated the purification capacity of bioreactor to low C / N urban sewage, Synthetic low C/N ratio wastewater prepared with tap water was used in this experiment. Ingredients included $C_6H_{12}O_6$(carbon source, 110 mg COD/L); $NH_4Cl$(source of $NH_3$-N,25mg N/L); $NaH_2PO_4$(source of P, 1.15mg P/L). At the same time, 1.0ml/L trace elements concentrate was added to the domestic wastewater [24]. The trace element concentrate is composed of: EDTA 5000 mg/L, $FeCl_2$ 3000 mg/L, $MnCl_2 \cdot 4H_2O$ 1000mg/L, $ZnCl_2 \cdot 7H_2O$ 450 mg/L, $CuCl_2 \cdot 5H_2O$ 270 mg/L, $NiCl_2 \cdot 6H_2O$ 200 mg/L, $CoCl_2 \cdot 6H_2O$ 240 mg/L.

## 2.3 Chemical analysis

All samples of the water were filtered through 0.45μm filter paper before analysis. COD concentration using COD rapid measurement analyzer (DR1010,HACH, USA). Ammonia nitrogen ($NH_4^+$-N) was measured by ammonia nitrogen tester (5B-3N, LianHua, China). The nitrite nitrogen ($NO_2^-$-N), nitro nitrogen ($NO_3^-$-N), total, Sulfate ($SO_4^{2-}$-S) and the sulfide ($S^{2-}$-S) was measured using a multi-parameter analyzer (DR3900, HACH, USA). The WTW real-time water quality testing equipment (Multi 3420, WTW, Germany) was used to monitor the pH and temperature. All water samples were measured in triplicate.

## 2.4 Analysis of microbial diversity

The samples extracted from the reactor were utilized for microbial community analysis using the E.Z.N.A.® soil DNA kit (Omega Bio-tek, Norcross, GA, U.S.). The total DNA of the microbial community was extracted following the provided instructions. The 16s rRNA gene was amplified by polymerase chain reaction (PCR) using 515F (5′-GTGYCAGCMG CCGC GG TAA-3′) and 907R (5′-CCGYCAATTYMTTTRAGTTT-3′), and the raw data was obtained by sequencing on the Illumina Hiseq2500 sequencing platform. OTU clustering of sequences based to 97% similarity using UPARSE software (version 7.1). Species classification annotation for each sequence using RDP classifier (version 2.2), aligned to the Silva 16S rRNA database (version 138) and set an alignment threshold of 70%.

# 3. Results and discussion

## 3.1 COD removal performance

The low C/N ratio wastewater reactor based on sulfur electron acceptor operated continuously for 90 days and the reactor showed the good performance in the whole process of treating synthetic wastewater. The removal performance of COD in the low C/N ratio wastewater

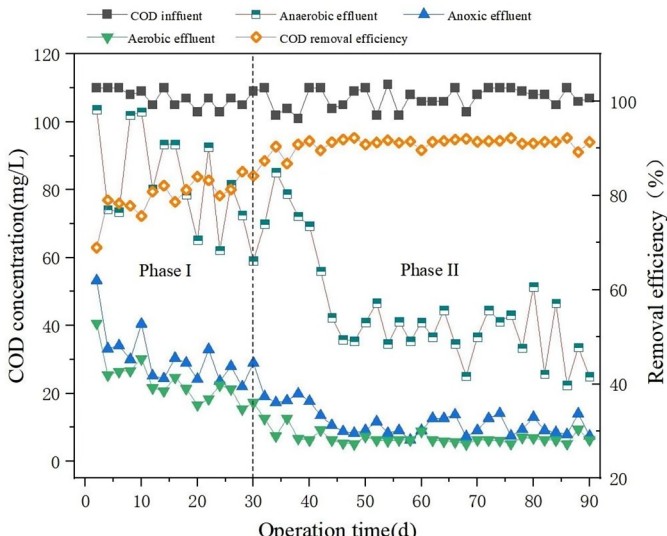

**Fig 2. COD concentration and removal efficiency of each reactor during 90 days.**

reactor are shown in Fig 2. The results demonstrated significant difference between Phase I and Phase II during the whole experiment, Phase II presented much better COD removal efficiency (94% ± 2.11%) than that of Phase I (78% ± 4.26%). Generally, the average effluent COD concentration of Phase II was 8.02±1.53 mg COD/L. In comparison, the average efflu-ent COD concentration of Phase I was 23.28±2.28 mg COD/L. The COD removal perfor-mance of Phase II is higher than the results of other studies with low C/N ratio wastewater treatment process [3] (C/N ratio = 4.821, average COD removal rate of 90%). the COD con-centration changes of each reactor at different phase were detected. Compared to the average COD concentration of anaerobic reactor (82.35mg COD/L) in the phase I, the average COD concentration of anaerobic reactor in the phase II is lower (44.33mg COD/L). The COD con-centration of the effluent from the anoxic reactor was close to that of the aerobic effluent, suggesting that the removal effect of COD after addition of the sulphur electron acceptor on was mainly in the anaerobic and anoxic reactor, which also suggests that a heterotrophic denitrification process may be taking place in the anoxic reactor using a small amount of car-bon source. In the phase II, the effluent COD concentration gradually decreased until it remained relatively stable. This may caused by the added sulfur electron acceptor changing the microbial community.

The higher COD removal rate in phase I (78% ± 4.26%)can be attributed to several factors. One reason is that the COD in synthetic sewage is composed of easily degradable glucose, and this COD removal rate is comparable to the COD removal rate observed by Wang under con-ditions of low C/N, suggesting a similarity in the composition of easily degradable carbon sources [25]. Additionally, heterotrophic bacteria play a crucial role by consuming a substan-tial amount of carbon sources through heterotrophic oxidation (Eq 1) and heterotrophic deni-trification (Eq 2) [26]. However, sulfate reduction metabolism process enhancing the removal of organic pollutants in wastewater during the phase II [27]. In this process, sulfate serves as an electron receptor in microbial metabolism, while glucose functions as an electron donor. This enables sulfate-reducing bacteria (SBR) to undergo substantial growth and reproduction, com-pleting the sulfate reduction metabolism process [28] (Eq 3). As a result, there is a

transformation in the original microbial community structure, as depicted in Figs 6 and 7.

$$1/24C_6H_{12}O_6 + 1/4O_2 \rightarrow 1/4CO_2 + 1/4H_2O$$
$$\Delta G = -120.1 kJ/mol$$

(1)

$$1/24C_6H_{12}O_6 + 1/5NO_3^- + 3/5H^+ \rightarrow 1/4CO_2 + 1/10N_2 + 7/20H_2O$$
$$\Delta G = -113.63 kJ/mol$$

(2)

$$1/24C_6H_{12}O_6 + 1/8SO_4^{2-} + 3/16H^+ \rightarrow 1/4CO_2 + 1/16H_2S + 1/16HS^- + 1/2H_2O$$
$$\Delta G = -20.69 kJ/mol$$

(3)

## 3.2 Nitrogen removal performance

Fig 3(a) and 3(b) illustrated the concentration changes of ammonia nitrogen, nitrate nitrogen, and nitrite nitrogen in each reactor throughout the experimental stage. As shown in Fig 3(a), the ammonia nitrogen removal concentration of the overall anaerobic reactor was low and changed little between the two phase, while the average effluent ammonia nitrogen concentration of the anoxic reactor in the phase I was reduced to 14.5±2.86mg N /L and that in the phase II was more significantly reduced (10.24±4.38mg N /L). This was may due to the reflux of denitrification liquid containing aerobic heterotrophic bacteria, denitrifiers, and nitrifying bacteria into the anoxic reactor. Additionally, a portion of the ammonia nitrogen in the anoxic reactor underwent conversion into nitrite nitrogen or nitrate nitrogen through nitrification reactions [29].

As shown in Fig 3(a), the effluent ammonia nitrogen of the aerobic reactor was also significantly different between the phase I and the phase II. Generally, the effluent ammonia nitrogen concentration of the aerobic reactor in Phase II was kept lower than 1.844 mg N / L after day 40. At the same time, the overall average ammonia nitrogen removal rate also increased from 73.92% to 92.12%. The overall results prove that the low-carbon and nitrogen ratio wastewater treatment process based on the sulfur electron acceptor can significantly remove the ammonia nitrogen from the wastewater with a low C/N ratio.

To assess the impact of adding a sulfur electron acceptor on the denitrification and nitrification reactions in the reactor, the dynamic changes in concentrations and removal rates of $NO_3^-$ and $NO_2^-$ at each phase were monitored. As shown in Fig 3(b), there was a slight variation in $NO_3^-$ in the effluent of the anaerobic reactor, possibly indicating that a small amount of sludge carrying $NO_3^-$ from the aerobic reactor was flow back into the anaerobic reactor.

In addition, the effluent $NO_3^-$ concentration of aerobic reactor in phase I fluctuates over time, while the average effluent $NO_3^-$ concentration of the anoxic reactor shows a slight reduction in phase I (5.17mgN/L to 3.33mgN/L), and the average effluent $NO_2^-$ increased slightly (0.91mgN/L to 1.33mgN/L),consistent with a previous study [30]. This indicates that heterotrophic denitrification occurs in the microbial community within the anoxic reactor (Eq 4) [31] and autotrophic nitrification(Eq 5) [32]. However, due to the limitation of carbon sources, the activity of heterotrophic denitrifiers was inhibited, resulting in lower denitrification efficiency of the reactor under conditions of a low carbon and nitrogen ratio in the influent [33].

$$1/24C_6H_{12}O_6 + 1/5NO_3^- + 3/5H^+ \rightarrow 1/4CO_2 + 1/10N_2 + 7/20H_2O$$
$$\Delta G = -113.63 kJ/mol$$

(4)

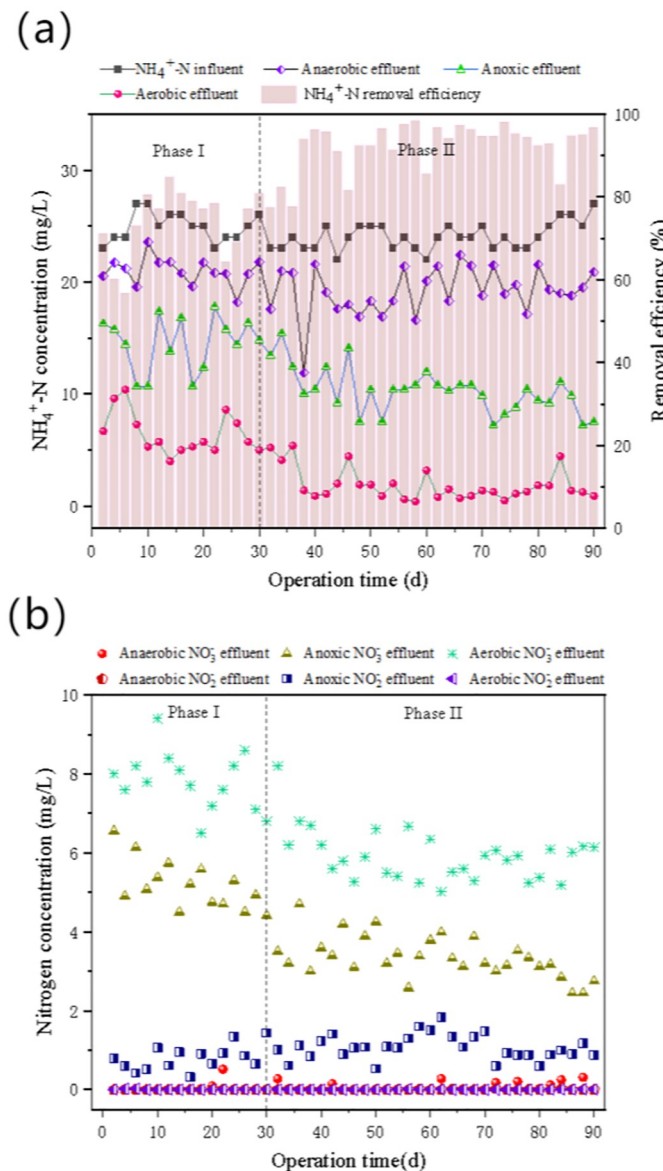

**Fig 3. Changes of ammonia nitrogen, nitrate nitrogen and nitrite nitrogen in each reactor effluent; (A)Changes in ammonia nitrogen concentration and removal efficiency; (B) Changes in nitrate nitrogen and nitrite nitrogen concentration.**

$$1/8NH_4 + +1/4O_2 \rightarrow 1/8NO_3^- + 1/4H^+ + 1/8H_2O$$

$$\Delta G = -43.64kJ/mol$$

(5)

Compared to the low nitrogen efficiency in phase I, the denitrification rate in phase II increased due to changes in the sulfur electron acceptors and the microbial community. As shown in Fig 3(b), the average $NO_3^-$ concentration in the aerobic effluent continuously decreased from 7.83 mgN/L to 5.92 mgN/L in phase I. In the phase II, the suppressed activity

of heterotrophic denitrifying bacteria may have resulted in an inability to fully perform denitrification. Meanwhile, autotrophic denitrifying bacteria, stimulated by the sulfur electron receptor, experienced significant growth and reproduction. This possibly led to the collaborative interaction between heterotrophic denitrifying bacteria and autotrophic denitrifying bacteria within the anoxic reactor, ultimately promoting the denitrification process in the reactor (Eq 6). This observation aligns with previous findings [34]. As the phase II progresses, the average $NO_2^-$ concentration in the anoxic effluent showed a declining trend by day 60, possibly due to the enhanced activity of sulfur autotrophic denitrifying bacteria (Fig 9).

$$3\,SO_4^{2-} + 4\,NH_4^+ \rightarrow 3\,S^{2-} + 4\,NO_2^- + 4\,H_2O + 8\,H^+$$
$$\Delta G = -47.8 \text{kJ/mol}$$
(6)

In contrast to conventional heterotrophic denitrification, thioautotrophic denitrifiers could utilize reducing sulfur compounds as electron donors and grow in a chemoautotrophic manner [35] (Eq 7). After the addition of the sulfur electron receptor, sulfate-reducing bacteria multiply and produce a large number of reducing sulfur compounds, leading to a rapid increase in the concentration of these compounds in the water. This, in turn, promotes the activity of sulfur autotrophic bacteria in the anoxic reactor, allowing microorganisms to maintain high nitrogen removal efficiency even in the presence of low carbon sources [5]. The sulfide concentration (Fig 5) and its identification in the sulfate-reducing reactor (Fig 9) support this observation. Moreover, from an electron transport perspective, the complete oxidation of sulfide or thiosulfate to sulfate is one of the most significant processes for electrochemical autotrophs. Therefore, autotrophic denitrification driven by reducing sulfur compounds holds great ecological significance [36].

$$1/16H_2S + 1/16HS^- + 1/5NO^{3-} + 1/80H^+ \rightarrow 1/8SO_4^{2-} + 1/10N_2 + 1/10H_2O$$
$$\Delta G = -92.94 \text{kJ/mol}$$
(7)

## 3.3 Phosphorus removal performance

The total phosphorus removal performance of the sulfur electron acceptor reactor for low C/N ratio wastewater was depicted in Fig 4. The reactor maintained a total phosphorus concentration of approximately 1.13 mg P/L. In the anaerobic reactor, the average total phosphorus concentration in phase I was 3.06 mg P/L and steadily increased. There was a notable surge in total phosphorus in 10 days before the start of the phase II, peaking at 38 days with a total phosphorus concentration of 5.4 mg P/L. However, as the reactor continued to operate, the average phosphorus concentration of anaerobic reactor in phase II was consistently higher than that in the phase I. It means that more and more phosphorus-accumulating organisms (PAO) were produced in the anaerobic reactor after the addition of sodium sulphate.

In the anaerobic phase, polyphosphates accumulate in biological PAO that absorb volatile fatty acids (VFA) from wastewater and store them to form intracellular carbon and energy substrate-polyhydroxy fatty acid ester (PHA) [37]. PAO cleaved the cellular polyphosphates to generate the energy needed by the substrate to release phosphorus [38]. In the anaerobic reactor, the phase II exhibited a more pronounced phosphate release phenomenon, indicating that volatile fatty acids (VFA) were heavily absorbed by the PAO [39]. Simultaneously, the sulfate reduction reaction in the anaerobic reactor produced a certain amount of sulfide, and a specific concentration of sulfide had a slight toxicity to microorganisms [40].

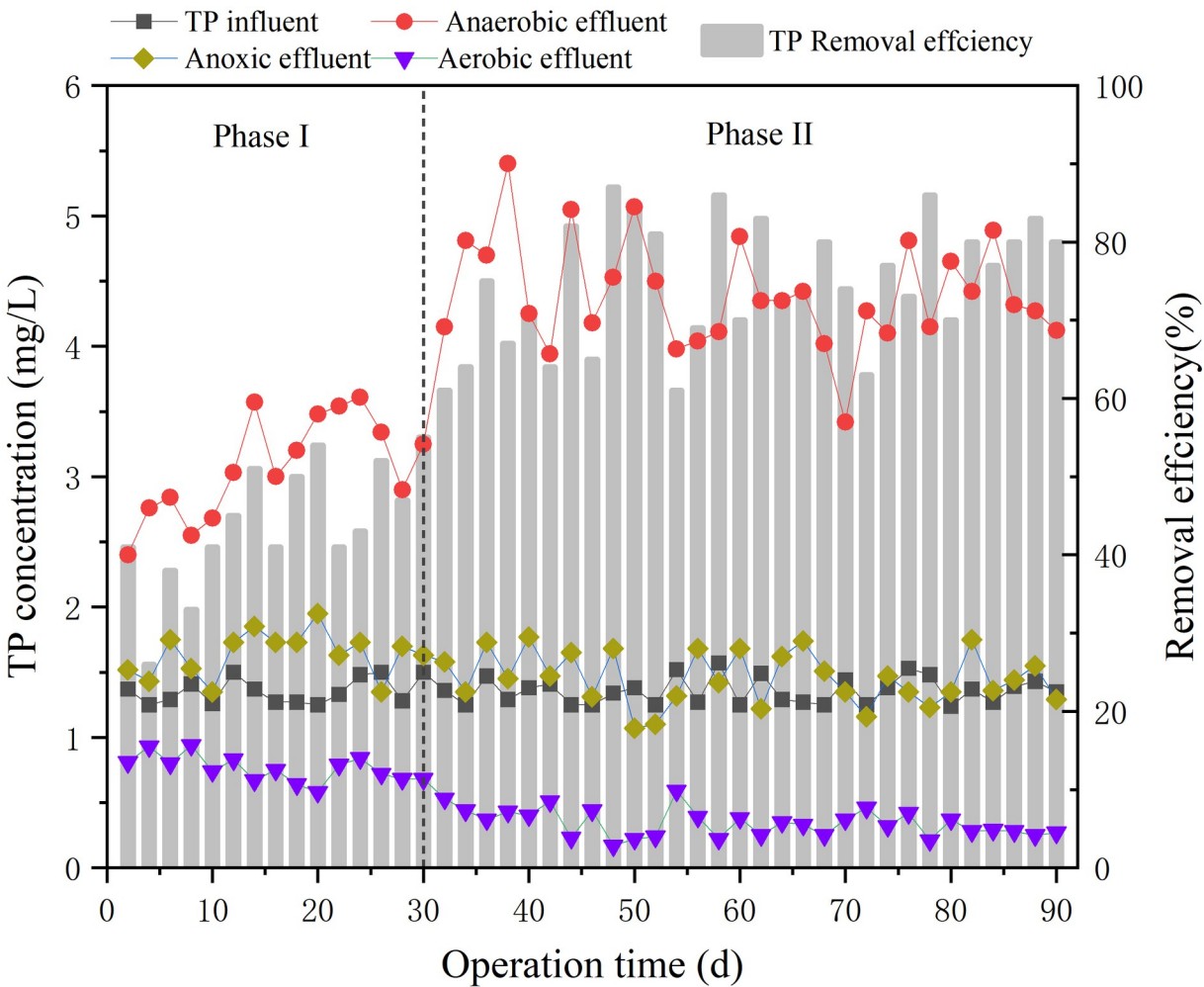

**Fig 4. Total phosphorus concentration and removal efficiency of each reactor during 90 days.**

According to studies, in an environment with higher concentrations of sulfide, microorganisms undergo an energy-consuming process of self-detoxification [41]. In this detoxification process, hydrogen ions are removed from the sulfide. The extracted hydrogen ions then generate a proton motive force to enhance the absorption of the driving VFA [42]. Therefore, an increase in a certain concentration of sulfide will result in higher PMF, more VFA absorption, and PHA storage, ultimately leading to a rapid increase in the total phosphorus concentration of the anaerobic reactor.

The phosphate concentration of anaerobic reactor was maintained at a certain concentration after 50 days of the experiment probably due to the fact that the accumulation of polyphosphate in the sludge has reached the upper limit [43]. Meanwhile, as shown in Fig 4, the overall concentration of total phosphorus in phase II (0.35 mg P/L) was lower than that of phase I (0.76 mg P/L), and the overall removal rate of total phosphorus in phase II (74%) was much higher than that of phase I(43%). This difference may be attributed to the production of more PHA in the anaerobic phase, which enhances the microbial phosphorus absorption during the aerobic phase.

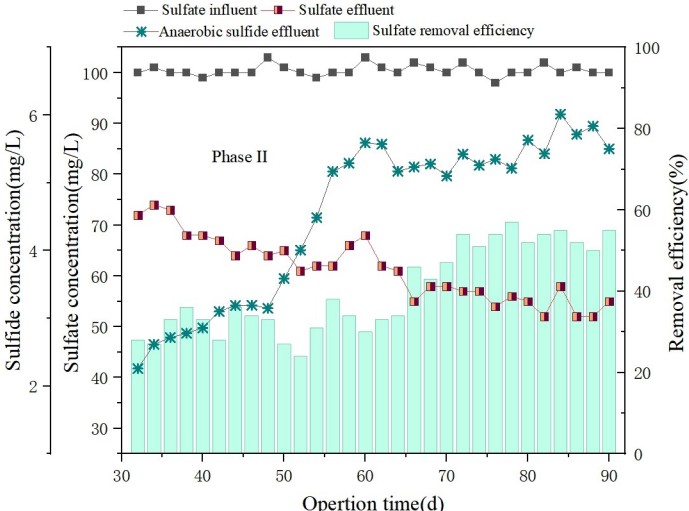

**Fig 5. Changes in anaerobic sulfide and overall effluent sulfate during phase II.**

## 3.4 Sulfate and sulfide transformation

The concentration of sulfate and the sulfide are not only important indicators of sulfur electron receptor transfer technology, but also reflect the activity of specific microorganisms, particularly sulfate-reducing bacteria. The effluent concentration changes of sulfate concentration and sulfide concentration after the addition of the sulfur electron acceptor was shown in Fig 5.

Since organic sulfur was not present in the synthetic sewage in this study, and sulfate and sulfide were relatively more stable in biological sulfur conversion than other forms of sulfur [44], so the main focus of this experiment was on the sulfate and sulfide. The overall effluent sulfate concentration was decreased to 50 mg S/L and the overall sulfate removal rate of reactor eventually stabilized at 70% during the whole experiment. Additionally, the sulfide concentration in anaerobic effluent gradually increased and finally stabilized at about 5 mg S /L. The result indicated that effective conversion of sulfate by sulfate-reducing bacteria through electron transfer into other sulfur sulfide. The organic matter in the water is transferred to sulfate by anaerobic bacteria in the anaerobic reactor (Eq 3). Simultaneously, some of the sulfide generated will couple with metal ions in the wastewater and sludge, forming precipitated sulfide, thereby reducing sulfide and metal ion concentration in the effluent. This process alleviates the burden on urban sewage treatment [45, 46]. The low sulfide concentration from sulfate conversion may due to the glucose-rich nature of the wastewater, which was easily decomposed and had a low C/N ratio.

## 3.5 Microbial community analysis

**3.5.1 Microbial community structure and diversity.** In order to investigated the mechanism of nitrogen and phosphorus removal in urban low C/N ratio wastewater, 16s rRNA sequencing technology was used to analyze the microbial community in each reactor. Microbial community samples were collected from the sludge in anaerobic, anoxic and aerobic reactor. The two experimental phases were marked as follows: AN (anaerobic—no sulfur electron acceptor), AN1 (anoxic—no sulfur electron acceptor), O (aerobic—no sulfur electron acceptor), and AN-S (anaerobic—added sulfur electron receptor), AN1-S (anoxic—added sulfur electron acceptor), O-S (aerobic—added sulfur electron acceptor).

**Table 1. Annotation and diversity analysis of the microbial communities for each sample.**

| Sample | Sequence | OTU | Coverage | Ace | Chao | Shannon | Simpson |
|--------|----------|-----|----------|-----|------|---------|---------|
| AN | 27731 | 3538 | 0.990 | 1219.12 | 1249.00 | 4.72 | 0.03 |
| AN1 | 35244 | 1539 | 0.991 | 1235.17 | 1195.02 | 3.18 | 0.25 |
| O | 30624 | 988 | 0.994 | 814.65 | 785.78 | 4.09 | 0.06 |
| AN-S | 39640 | 34101 | 0.996 | 721.09 | 693.06 | 1.86 | 0.48 |
| AN1-S | 34000 | 34000 | 0.990 | 1336.34 | 1322.53 | 3.99 | 0.10 |
| O-S | 34101 | 39640 | 0.993 | 1212.25 | 1179.13 | 4.86 | 0.02 |

As shown in Table 1, the number of efficient operational taxon unit (OTU) sequences in the microbial community increased following the addition of sulfur electron acceptors. The microbial library coverage of the six biological samples exceeded 99%, indicating that the results comprehensively encompassed all microorganisms in the samples, further affirming the authenticity of the experiment.

Venn diagram was utilized analyze the distribution of OTU before and after the treatment, as shown in Fig 6. The microbial communities in the three reactors exhibited high abundance, particularly between anaerobic and anoxic sludge. However, substantial changes were observed in the anoxic and aerobic reactors, decreasing from 357 to 270 and 225 to 140, respectively. Simultaneously, the number of shared species among the three reactors decreased from 206 to 156, and the common species between the two reactors also decreased. Conversely, the number of unique species in each reactor increased, leading to a reduction in microbial community similarity between the reactors. These data suggest that, each reactor gradually developed its own dominant microbial colonies after the addition of the electron receptor.

As shown in Table 2, the diversity index(CHAO, Shannon, Simpson) of the biological community of the anaerobic sludge decreased after the addition of sulfur electrons. On the contrary, the diversity index of anoxic and aerobic sludge increased after the treatment, but these did not affect the increase of the total amount of microbial community species in the sludge.

The dilution curve primarily utilized the microbial Alpha diversity index of each sample at different sequencing depths to construct the curve, reflecting the microbial diversity of each sample at varying sequencing numbers. Fig 7(a) illustrated the dilution curves of the sludge samples in the three different reactors. The dilution curves increase with the OTU until they eventually become flat. The order of community diversity in the samples was AN-1-S > AN-S > O-S. Fig 7(b) showed the degree of difference in the abundance distribution of the species between the samples, as well as the quantified distance matrix. In the order of AN-S and O-S > AN1-S and O-S > AN-S and AN1-S.

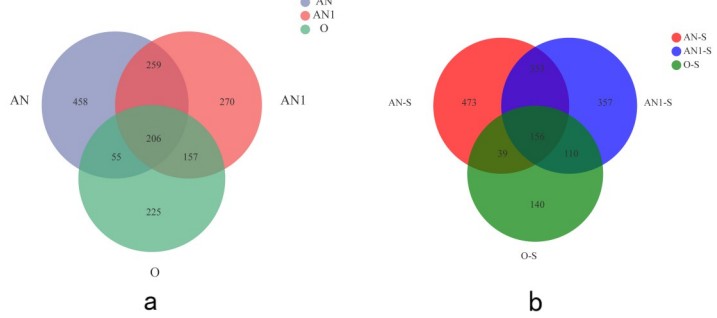

**Fig 6. OTU diagram of each reactor (a) Phase I (b) Phase II.**

**Table 2. The experimental parameters of phases I and II.**

| Parameter | Phase I (Day 1–30) | Phase II (Day 31–90) |
|---|---|---|
| C:N ratio | 4.4 | 4.4 |
| SRT(h) | 12 | 12 |
| NH4 $^+$(mg/L) | 25 | 25 |
| SO$_4$$^{2-}$ (mg/L) | 0 | 100 |
| HRT | 20 | 20 |

Fig 7(c) illustrated the changes in the rank-abundance curve. Among the three samples, the curve of AN-S and AN1-S sludge was relatively smooth and had the broadest coverage, indicating the highest species richness and biological population uniformity in the anoxic sludge. The second most prominent is the O-S sludge. There was no clear difference between AN-S and AN1-S. Under either AN-S or AN1-S conditions, the largest number of microbial species were present, and the species were evenly distributed. AN1-S was influenced by two inflows: one was the effluent of AN-S, and the other was the reflux of O-S, making it the most abundant. However, the anaerobic area was only influenced by the original wastewater.

Principal component analysis (PCA) was also used to assess differences between microbial communities. Fig 7(d) showed the PCA results for the different reaction zones, which indicated the differences in the microbial composition between the three reaction zones. The results were consistent with the results expressed in grade-scale Fig 7(c), and the microbial composition was different under different treatment conditions, not due to the growth of microorganisms themselves, which indicated that environmental factors had an important influence on the distribution of microorganisms.

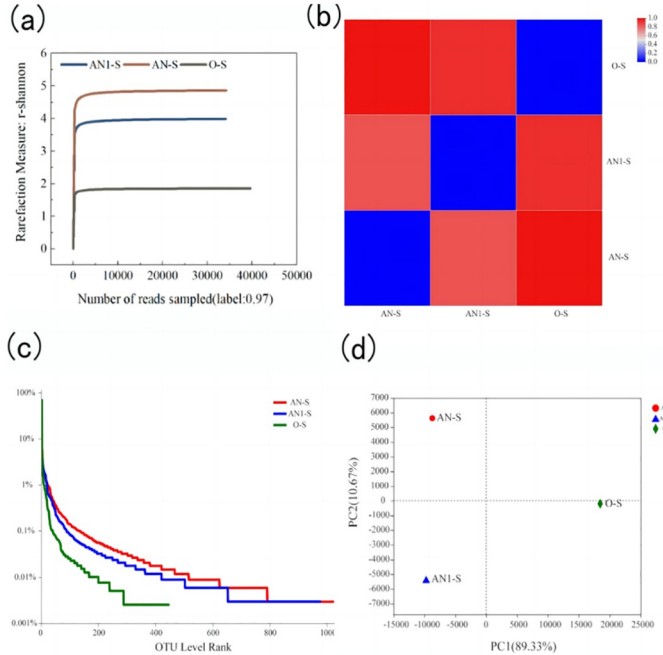

**Fig 7. Diversity and diversity analysis of microbial communities: (a) Shannon dilution curve (b) distance matrix Heatmap plot (c) abundance curve (d) principal component analysis.**

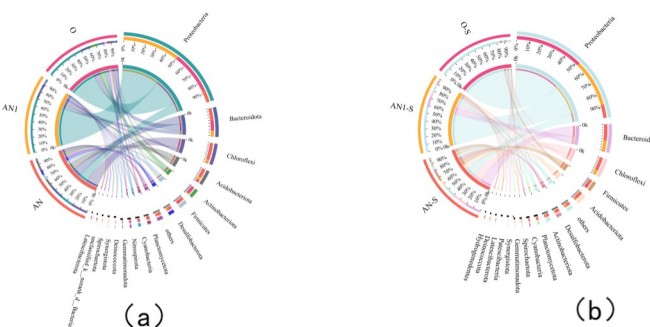

**Fig 8. Phylum-level microbial community changes in each reactor (a) Phase I (b) Phase II.**

**3.5.2 Analysis of microbial community diversity.** In order to further analyzed the core community and the major functional bacteria, the six sludge samples were isolated before and after the treatment, and the results showed that the microbial community had changed greatly (Fig 8). Proteus *Proteobacteria* (47.1%), *Bacteroidetes Bacteroidota* (12.5%), *Chloroflexi* (9.2%) *Acidobacteriota* (6.8%), and *Bacteroidetes Actinobacteriota* (4.3%) were the top five bacterial phyla in the samples before the addition of sulfur electron acceptors. After the addition of sulfur electron acceptors for 60 days, the bacterial abundance of *Proteobacteria* and *Firmicutes* increased to 54.1% and 4.1%, respectively. These phyla were reported to be enriched in some known sulfate-reducing species [47, 48], and their abundance increased due to the continuous application of electron acceptors. Another factor contributing to the relatively rapid increase in *Proteobacteria* may be the enrichment of some denitrifiers in *Proteobacteria* after the impact of the low C/N ratio [5]. However, the relative abundance of *Bacteroidota*, *Chloroflexi*, *Acidobacteriota*, and *Actinobacteriota* decreased to 12.0%, 8.2%, 4.6%, and 2.6%, respectively. These changes aligned with previous experiments involving elemental sulfur [49].

The top 50 genera were identified in samples after the addition of sulfur electrons to analyzed the microbial function of the sulfur electron acceptor technology during wastewater treatment. The genus-level heat maps of AN-S, AN1-S, and O-S were shown in Fig 9. It was clearly proved from the figure that the biological community of O-S had low species similarity with AN-S and AN-1, and the genus *Rhodanobacter* had an extremely high relative abundance (74.51%) in the O-S sludge. *Rhodobacter* which belonged to the γ -*Proteobacteria* was a part of the *xanthomonas* family within the broader category of *Proteobacteria*. Some studies had shown that this genus is associated with ammonification, ammonia assimilation, nitrogen fixation, and denitrification pathways [50]. The strains of this genus had been found to efficiently remove inorganic triplet nitrogen in the presence of high concentrations of organic nitrogen compounds [51].

Meanwhile, low concentrations of nitronitrogen and high nitrate in O-S promoted the growth of *Rhodobacter sp* strain [52]. This was attributed to the strong selective pressure on bacterial communities in environments with high nitrate contamination [53], leading to a reduction in microbial diversity and favoring the dominance of the species *Rhodobacter*. This aligned with previous denitrification data on water quality, suggesting that the enrichment of *Rhodobacter* was effective in removing nitrate and nitrate from water [53]. However, other significant genera in O-S included *Zoogloea* (1.04%), *Obscuribacteraceae* (2.43%), and *Micropepsaceae* (1.61%). The presence of Folium genus may contributed to an increase in EPS, promoting bacterial and nutrient aggregation in low carbon and nitrogen ratio environments, ultimately enhancing nitrogen removal efficiency and sedimentation [25, 54].

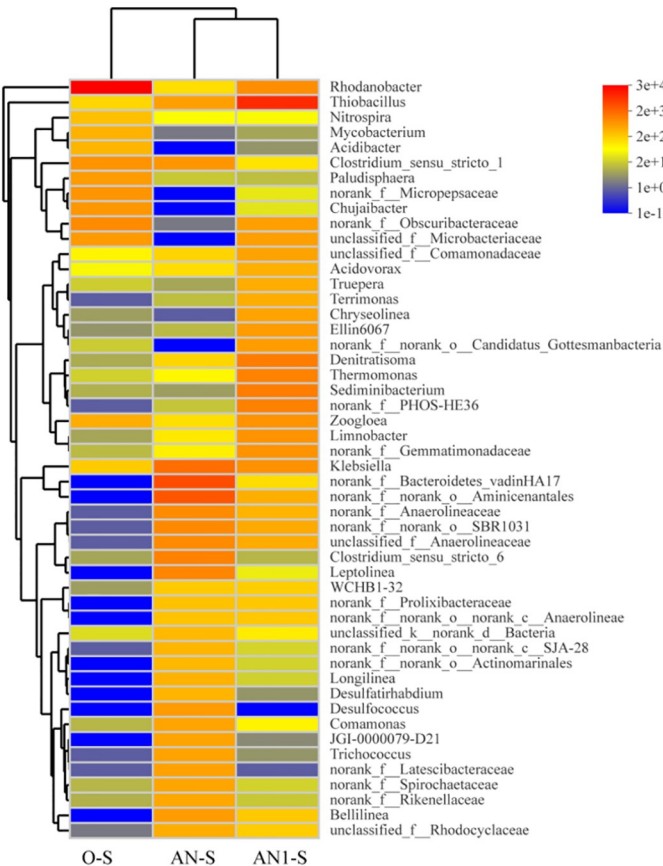

**Fig 9. The genus horizontal heat map of each reactor after the addition of the sulfur electron acceptor.**

The species and relative abundance of AN-S and AN1-S were similar, but there still were some differences. The main genera in AN-S were Bacteroidetes-vadinHA17 (13.04%), *Aobacterium Aminicenantales* (10.16%), *Leptolinea* (3.14%), and *Klebsiella* (5.68%). However, the primary bacterial genera in AN1-S were *Thiobacillus* (30.05%), *Denitratisoma* (3.88%), *PHOS-HE36* (3.63%), and *Thermomonas* (3.46%). It had been reported that *Bacteroides vadinHA17* can achieve carbon and nitrogen reduction in collaboration with *Denitratisoma*, *Clostridium*, and *Anaerolineaceae* without the addition of additional organic carbon sources [55]. Therefore, the addition of sulfur electron acceptors may also enhanced this cooperative reduction, leading to nitrogen and phosphorus removal. *Aminicenantes* function as anaerobic organic nutrient decomposers, fermenting carbohydrates and protein substrates, and respiring with nitrite. They primarily act as decomposers of organic matter, producing hydrogen and acetic acid [56]. This suggested that *Aminicenantes* can efficiently utilize sugar-based organic matter in wastewater, conducting fermentation to enhance bioavailability in conditions of low C/N ratios.

In AN1-S sludge, *Thiobacillus* (30.05%) emerged as the absolute dominant genus, potentially serving as a key factor in the efficient nitrogen removal process in the AN1-S reactor. *Thiobacillus* comprised mainly obligate autotrophs, with *Thiobacillus denitrificans* being a model species within this genus. While *Thiobacillus* oxidizes hydrogen sulfide, elemental sulfur, and *Thiosulfate* to sulfate in water, it also interacts with other microorganisms in water,

including endogenous denitrifying bacteria like *Denitratisoma* and highly active denitrifiers such as *Thermomonas* [57]. This interaction provided a substantial number of electrons, fostering the growth and reproduction of these denitrifiers and promoting the denitrification rate in the AN1-S reactor. Additionally, it had been reported that PHOS-HE36 was not only present in highly efficient denitrification communities, but it may also contributed to phosphorus accumulation, enhanced phosphate enrichment in the ecosystem and further reduced total phosphorus concentration in the water [58]. These findings aligned with the observed changes in wastewater effluent quality [59]. In summary, the addition of a specific amount of sulfur electron acceptor induced changes in the microbial community of low C/N ratio wastewater. The dominance of certain strains and obligate autotrophic species enhanced the synergistic effect, leading to enhanced removal of excessive nitrogen and phosphorus from the wastewater.

### 3.6 Shortcoming and potential of sulfur electron acceptor process

This research primarily focused on the characteristics of the low C/N ratio in urban sewage treatment plants, utilizing sulfate as the sole electron acceptor to facilitate a rapid sulfur reduction process. Upon comparing the changes in pollutant removal concentration before and after the addition, it became evident that the efficiency of nitrogen and phosphorus removal post the introduction of sulfur electron acceptors surpasses that of traditional sewage treatment processes. Moreover, sulfate was more cost-effective than the added carbon source, so the operational costs in practical use would be lower than those in the latter. These data suggests that treating urban low-N ratio wastewater through the addition of sulfur electron acceptors is an economically feasible method.

While the addition of a sulfur electron receptor to the reactor enhances phosphorus removal efficiency in low C/N ratio wastewater, it introduced the challenge of hydrogen sulfide gas production. Additionally, after 60 days of addition, only 50% of the sulfate concentration has been reduced, leaving the remaining 50% in the sewage and sludge. This situation may lead to elevated effluent sulfate levels, necessitating further treatment steps. The availability of sulfur sources and the potential toxicity of high sulfate and hydrogen sulfide to other functional bacteria were also needed to consider. Despite these challenges, the technology still hold promise for treating urban wastewater with a low C/N ratio. Addressing these concerns could involve adjusting the proportion of sulfur sources in urban wastewater, modifying the carbon-nitrogen ratio and sulfur-nitrogen ratio, hydraulic retention time, or reflux mode to control sulfate concentration and sulfide formation in the water. Furthermore, the application of sulfur electron receptor technology in industrial wastewater, such as mine wastewater and heavy metal wastewater, presented significant practical potential [60–62]. For instance, coupling hydrogen sulfide and metal ions in wastewater could recycled metal sulfide and reduced by-product costs in the wastewater treatment process.

### 4. Conclusions

The sulfur electron acceptor(sulfate) was added into the traditional low C/N wastewater treatment significantly could enhanced the nitrogen and phosphorus removal processes in biological treatment. This was due to the unique sulfur microorganisms were cultivated in each bioreactor, achieving efficient removal of $NH_4^+$-N、$NO_2^{2-}$N、$NO_3^{2-}$N, TP, and COD. Simultaneously, the residual rate of sulfate electron acceptor treatment was only 50%, and the remaining sulfate can be control through additional steps.

The metagenome and community diversity of bacterial communities after the addition of sulfur electron receptors revealed the enrichment of distinctive functional microorganisms in

each reactor, such as *Proteobacteria*, *Firmicutes*, and *Chlorobacteria*, and genera like *Rhodobacterium*, *Deshibacter*, and *Bacteroides*. These microorganisms may played a crucial role in enhancing pollutant removal in low C/N ratio wastewater.

Although the application of sulfur electron acceptors in low C/N ratio wastewater processes poses challenges like sulfide production and excessively high residual sulfate concentration, it undeniably proves effective in treating low C/N ratio wastewater. Future research should delve into understanding the potential mechanisms of dominant strains under low carbon stress, the secondary utilization of residual sulfuric acid, the phylogenetic characteristics of nitrogen and phosphorus removal microorganisms, and the development of new sulfur electron acceptors suitable for low C/N ratios.

## Author Contributions

**Conceptualization:** Li Wei.

**Data curation:** Erming Luo, Xinxin Zhang, Qian Lu, Dong Wei, Chun ying Li.

**Formal analysis:** Jia Ouyang, Qian Lu.

**Investigation:** Erming Luo, Yongcheng Wang.

**Methodology:** Erming Luo, Dong Wei.

**Project administration:** Jia Ouyang.

**Resources:** Xinxin Zhang, Dong Wei, Yongcheng Wang, Zhengjiong Cha, Chengwei Ye.

**Supervision:** Chun ying Li, Li Wei.

**Writing – original draft:** Erming Luo.

**Writing – review & editing:** Li Wei.

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
