## [Decision Letter · Decision Letter 0]

4 Jun 2024

PONE-D-24-01884Study on the Enhancement of Low Carbon-to-Nitrogen Ratio Urban Wastewater Pollutant Removal Efficiency by Adding Sulfur Electron AcceptorsPLOS ONE

Dear Dr. Wei,

Thank you for submitting your manuscript to PLOS ONE. After careful consideration, we feel that it has merit but does not fully meet PLOS ONE’s publication criteria as it currently stands. Therefore, we invite you to submit a revised version of the manuscript that addresses the points raised during the review process.

 Be sure to:Indicate which changes you require for acceptance versus which changes you recommendAddress any conflicts between the reviews so that it's clear which advice the authors should followProvide specific feedback from your evaluation of the manuscript==============================

We look forward to receiving your revised manuscript.

Kind regards,

Guanglei Qiu

Academic Editor

PLOS ONE

Journal Requirements:

[This study was supported by the Basic and Applied Basic Research Project of the Natural Science Foundation of Heilongjiang Province of China（Grant No.LH2019E068），the Guangzhou Municipal Science and Technology Project(Grant No.202201011743; Grant No.202201011683; Grant No.202201011584；Grant No.2024A03J0392), the International Science and Technology Cooperation Project of the Guangzhou Economic and Technological Development Zone (Grant No.2020GH04), and the zengcheng Innovation and Entrepreneurship Support Program for Leading Talents of Guangzhou City (Grant No.2017012), the livelihood science and technology project of Nansha District, Guangzhou (Grant No.2021MS017).]

 [The author(s) received no specific funding for this work.]

4. We note that your Data Availability Statement is currently as follows: [All relevant data are within the manuscript and its Supporting Information files]

Reviewers' comments:

Reviewer's Responses to Questions

**Comments to the Author**

1. Is the manuscript technically sound, and do the data support the conclusions?

Reviewer #1: Partly

Reviewer #2: Yes

2. Has the statistical analysis been performed appropriately and rigorously? 

Reviewer #1: No

Reviewer #2: No

3. Have the authors made all data underlying the findings in their manuscript fully available?

Reviewer #1: Yes

Reviewer #2: Yes

4. Is the manuscript presented in an intelligible fashion and written in standard English?

Reviewer #1: No

Reviewer #2: Yes

5. Review Comments to the Author

Reviewer #1: Manuscript Number: PONE-D-24-01884

Title: Study on the Enhancement of Low Carbon-to-Nitrogen Ratio Urban Wastewater Pollutant Removal Efficiency by Adding Sulfur Electron Acceptors

Comment 1: The abstract part seems shallow; not addressed the main points of the study.

Comment 2: in introduction part, you provided general issues about C/N. its better if you concentrated on the reality of your study areas. The introduction part also required to summarize more than this.

Comment 3: on line 138, you stated figure 1. But there wasn’t it

Comment 4: Your research design and methodology the way of sample collection and preservation needs modification

Comment 5: the style you wrote the equations not followed the standard

Comment 6: your result description wasn’t smart. It needs modification

Comment 7: would you followed the reference writing style of PLOSE ONE?

Reviewer #2: This study evaluated the removal efficiency of characteristic pollutants during wastewater treatment before and after using sodium sulfate as an electron acceptor in a simulated urban wastewater treatment plant (WWTP) bioreactor. However, the manuscript was not clearly illustrated, and some information is missing. The specific comments and suggestions are as follows.

1. The figures in this manuscript seems very blur.

2. What is the organic matter used for the synthetic wastewater preparation?

3. What is the “The necessary trace elements” added to the domestic wastewater?

4. In my opinion, it is better to add a table to show the operational parameters for phase I and II.

5. What is the meaning of “dynamic changes”?

6. The writing English should be improved.

7. Why did the authors add polyurethane fillers to the aerobic reactor?

6. PLOS authors have the option to publish the peer review history of their article (what does this mean?). If published, this will include your full peer review and any attached files.

Reviewer #1: No

Reviewer #2: No

---

## [Author Response · Author response to Decision Letter 0]

24 Jul 2024

Response to Reviewers

Dear Editors and Reviewers,

The enclosed manuscript entitled “Study on the Enhancement of Low Carbon-to-Nitrogen Ratio Urban Wastewater Pollutant Removal Efficiency by Adding Sulfur Electron Acceptors” is a revised submission (Manuscript Number: PONE-D-24-01884). Many thanks to the editor and reviewers for their valuable comments on the paper. So we have made substantial changes to it, including the language and writing, the introduction, and logical description. All the authors wish it could be considered for publication in PLOS ONE. 

The comments from the editor/reviewers and the response to editor/reviewers are described as follows, 

Journal Requirements: 

1.When submitting your revision, we need you to address these additional requirements.

Reply: OK，additional requirements has been addressed rightly

2＆3.We note that the grant information you provided in the ‘Funding Information’ and ‘Financial Disclosure’ sections do not match. funding information should not appear in the Acknowledgments section or other areas of your manuscript

Reply:Funding Information has been removed from the manuscript and update in cover letter.

4.We note that your Data Availability Statement is currently as follows: [All relevant data are within the manuscript and its Supporting Information files]

Reply:yes , I confirmed that my submission contains all raw data required to replicate the results of your study

Reviewer #1: 

1) The abstract part seems shallow; not addressed the main points of the study.

Reply: 

→ ‘We had modified the Abstract to highlight the purpose of this article, the experimental approach, the results and future research..’ 

2) In introduction part, you provided general issues about C/N. its better if you concentrated on the reality of your study areas. The introduction part also required to summarize more than this. 

Reply: 

→ the reality of your study areas has been Added into the introduction and the logical description in introduction had been modified . 

3) on line 138, you stated figure 1. But there wasn’t it

Reply: 

→ figure 1 was added on the bottom of manuscript(line 883)

4) Your research design and methodology the way of sample collection and preservation needs modification

Reply: 

→ The design and methodology the way of sample collection and preservation has been revised in the manuscript. 

‘2.2 Inoculation sludge and synthetic sewage

In this study, the inoculated sludge came from a sewage treatment plant in Nansha, Guangzhou, which operated stably for a long time. Before the sludge was inoculated, the sludge was cleaned to remove impurities in the sludge. 5L of anaerobic sludge was inoculated in each of the two UASB bioreactors with a sludge concentration of about 4000 to 5500 mg / L. At the same time, 3L aerobic sludge was inoculated in the aerobic reactor and 30% polyurethane fillers was added to the aerobic sludge for mixing.

In order to investigated the purification capacity of bioreactor to low C / N urban sewage, Synthetic low C/N ratio wastewater prepared with tap water was used in this experiment. Ingredients included C6H12O6(carbon source, 110 mg COD/L); NH4Cl(source of NH3-N,25mg N/L); NaH2PO4(source of P, 1.15mg P/L). At the same time, 1.0ml/L trace elements concentrate was added to the domestic wastewater[24]. The trace element concentrate is composed of :EDTA 5000 mg/L，FeCl 2  3000 mg/L，MnCl2·4H2O 1000mg/L, ZnCl2·7H2O 450 mg/L, CuCl2 ·5H2O 270 mg/L， NiCl 2 ·6H 2 O 200 mg/L ， CoCl 2 ·6H 2 O 240 mg/L.

2.3 Chemical analysis

All samples of the water were filtered through 0.45μm filter paper before analysis. COD concentration using COD rapid measurement analyzer (DR1010,HACH, USA). Ammonia nitrogen (NH4+-N) was measured by ammonia nitrogen tester (5B-3N, LianHua, China). The nitrite nitrogen (NO2--N), nitro nitrogen (NO3--N), total , Sulfate (SO42--S) and the sulfide (S2--S) was measured using a multi-parameter analyzer (DR3900, HACH, USA). The WTW real-time water quality testing equipment (Multi 3420, WTW, Germany) was used to monitor the pH and temperature.All water samples were measured in triplicate.’.

5）the style you wrote the equations not followed the standard

Reply: 

→ The equations has been revised in the manuscript.For example

 （1）

ΔG= -120.1kJ/mol

6）your result description wasn’t smart. It needs modification

Reply: 

→ The logical description and structure of result has been revised in the manuscript.

7）would you followed the reference writing style of PLOSE ONE?

Reply: 

→ The reference writing style has been revised in the manuscript

1.Song H, Feng J, Zhang L, et al.Advanced treatment of low C/N ratio wastewater treatment plant effluent using a denitrification biological filter: Insight into the effect of medium particle size and hydraulic retention time.Environmental Technology & Innovation, 2021, 24: 102044.https://doi.org/10.1016/j.eti.2021.102044

2.Preisner M, Neverova-Dziopak E, Kowalewski Z. Analysis of eutrophication potential of municipal wastewater. Water Science and Technology, 2020, 81(9): 1994-2003.https://doi.org/10.2166/wst.2020.254

3.Li D, Li W, Zhang D, et al. Performance and mechanism of modified biological nutrient removal process in treating low carbon-to-nitrogen ratio wastewater. Bioresource Technology, 2023, 367: 128254. https://doi.org/10.1016/j.biortech.2022.128254

4.Yuan C, Zhao F, Zhao X, et al. Woodchips as sustained-release carbon source to enhance the nitrogen transformation of low C/N wastewater in a baffle subsurface flow constructed wetland. Chemical Engineering Journal, 2020, 392: 124840.https://doi.org/10.1016/j.cej.2020.124840

5.Huang W, Gong B, Wang Y, et al.Metagenomic analysis reveals enhanced nutrients removal from low C/N municipal wastewater in a pilot-scale modified AAO system coupling electrolysis.Water Research, 2020, 173: 115530.https://doi.org/10.1016/j.watres.2020.115530

6.Awasthi M K, Sarsaiya S, Wainaina S, et al. Techno-economics and life-cycle assessment of biological and thermochemical treatment of bio-waste. Renewable and Sustainable Energy Reviews, 2021, 144: 110837. https://doi.org/10.1016/j.rser.2021.110837

7.Duan S, Zhang Y, Zheng S. Heterotrophic nitrifying bacteria in wastewater biological nitrogen removal systems: A review.Critical Reviews in Environmental Science and Technology, 2022, 52(13): 2302-2338. https://doi.org/10.1080/10643389.2021.1877976

8.Wang B, Hu H, Huang S, et al.Simultaneous nitrate and sulfate biotransformation driven by different substrates: comparison of carbon sources and metabolic pathways at different C/N ratios.RSC Advances, 2023, 13(28): 19265-19275.

9.Wu L, Wei W, Xu J, et al.Denitrifying biofilm processes for wastewater treatment: developments and perspectives.Environmental Science: Water Research & Technology, 2021, 7(1): 40-67.https://doi.org/10.1039/D3RA02749J

10.Hellinga C, Schellen A, Mulder J W, et al. The SHARON process: an innovative method for nitrogen removal from ammonium-rich waste water. Water science and technology, 1998, 37(9): 135-142.

11.Jin R C, Yang G F, Yu J J, et al. The inhibition of the Anammox process: a review. Chemical engineering journal, 2012, 197: 67-79.

12.Lotti T, Kleerebezem R, van Erp Taalman Kip C, et al. Anammox growth on pretreated municipal wastewater. Environmental science & technology, 2014, 48(14): 7874-7880.

13.Van Dongen U, Jetten M S M, van Loosdrecht M C M. The SHARON®-Anammox® process for treatment of ammonium rich wastewater. Water science and technology, 2001, 44(1): 153-160.

14.Sousa A F S. The SHARON-anammox process for the treatment of ammonium-rich liquid residues produced by the anaerobic digestion of municipal solid wastes: a preliminary evaluation. Institute of Environmental Geology and Geoengineering, National Research Council, 2016.

15.Soliman M, Eldyasti A. Ammonia-Oxidizing Bacteria (AOB): opportunities and applications—a review. Reviews in Environmental Science and Bio/Technology, 2018, 17(2): 285-321.

16.Lotti T, Kleerebezem R, Lubello C, et al. Physiological and kinetic characterization of a suspended cell anammox culture. Water research, 2014, 60: 1-14.

17.Kumar U, Panneerselvam P, Gupta V V S R, et al. Diversity of sulfur-oxidizing and sulfur-reducing microbes in diverse ecosystems. Advances in Soil Microbiology: Recent Trends and Future Prospects: Volume 1: Soil-Microbe Interaction, 2018: 65-89.https://doi.org/10.1007/978-981-10-6178-3_4

18.Zhang L, Zhang S, Wang S, et al. Enhanced biological nutrient removal in a simultaneous fermentation, denitrification and phosphate removal reactor using primary sludge as internal carbon source. Chemosphere, 2013, 91(5): 635-640.

19.Nguyen T A, Juang R S. Treatment of waters and wastewaters containing sulfur dyes: A review. Chemical engineering journal, 2013, 219: 109-117.

20.Brown K A. Sulphur in the environment: a review. Environmental Pollution Series B, Chemical and Physical, 1982, 3(1): 47-80.

21.Selvasembian R, Gwenzi W, Chaukura N, et al. Recent advances in the polyurethane-based adsorbents for the decontamination of hazardous wastewater pollutants. Journal of Hazardous Materials, 2021, 417: 125960. https://doi.org/10.1016/j.jhazmat.2021.125960

22.Shao M F, Zhang T, Fang H H P. Sulfur-driven autotrophic denitrification: diversity, biochemistry, and engineering applications Applied microbiology and biotechnology, 2010, 88: 1027-1042.https://doi.org/10.1007/s00253-010-2847-1

23.Huang S, Yu D, Chen G, et al. Realization of nitrite accumulation in a sulfide-driven autotrophic denitrification process: Simultaneous nitrate and sulfur removal. Chemosphere, 2021, 278: 130413. https://doi.org/10.1016/j.chemosphere.2021.130413

24.Zhang Y, Zhang L, Li L, et al.A novel elemental sulfur reduction and sulfide oxidation integrated process for wastewater treatment and sulfur recycling.Chemical Engineering Journal, 2018, 342: 438-445. https://doi.org/10.1016/j.cej.2018.02.105

25.Li D, Li W, Zhang D, et al. Performance and mechanism of modified biological nutrient removal process in treating low carbon-to-nitrogen ratio wastewater. Bioresource Technology, 2023, 367: 128254. https://doi.org/10.1016/j.biortech.2022.128254

26.Wang X, Jiang C, Wang D, et al.Quorum sensing responses of activated sludge to free nitrous acid: Zoogloea deformation, AHL redistribution, and microbiota acclimatization.Water Research, 2023, 238: 119993. https://doi.org/10.1016/j.watres.2023.119993

27.Xu Z, Dai X, Chai X.Effect of different carbon sources on denitrification performance, microbial community structure and denitrification genes.Science of the Total Environment, 2018, 634: 195-204. https://doi.org/10.1016/j.scitotenv.2018.03.348

28.Wu D, Ekama G A, Chui H K, et al.Large-scale demonstration of the sulfate reduction autotrophic denitrification nitrification integrated (SANI®) process in saline sewage treatment.Water research, 2016, 100: 496-507. https://doi.org/10.1016/j.watres.2016.05.052

29.Liamleam W, Annachhatre A P.Electron donors for biological sulfate reduction.Biotechnology advances, 2007, 25(5): 452-463. https://doi.org/10.1016/j.biotechadv.2007.05.002

30.Sharp R, Khunjar W, Daly D, et al. Nitrogen removal from water resource recovery facilities using partial nitrification, denitratation-anaerobic ammonia oxidation (PANDA). Science of The Total Environment, 2020, 724: 138283. https://doi.org/10.1016/j.scitotenv.2020.138283

31.He Q, Cheng Z, Zhang D, et al.A sulfur-based cyclic denitrification filter for marine recirculating aquaculture systems.Bioresource Technology, 2020, 310: 123465.https://doi.org/10.1016/j.biortech.2020.123465

32.Wang H, Jiang C, Wang X, et al.Application of internal carbon source from sewage sludge: A vital measure to improve nitrogen removal efficiency of low C/N wastewater.Water, 2021, 13(17): 2338.https://doi.org/10.3390/w13172338

33.Di Capua F, Pirozzi F, Lens P N L, et al.Electron donors for autotrophic denitrification.Chemical Engineering Journal, 2019, 362: 922-937. https://doi.org/10.1016/j.cej.2019.01.069

34.Fu X, Hou R, Yang P, et al. Application of external carbon source in heterotrophic denitrification of domestic sewage: A review. Science of The Total Environment, 2022, 817: 153061.https://doi.org/10.1016/j.scitotenv.2022.153061

35.Liu F, Wang S, Zhang X, et al.Nitrate removal from actual wastewater by coupling sulfur-based autotrophic and heterotrophic denitrification under different influent concentrations.Water, 2021, 13(20): 2913.https://doi.org/10.3390/w13202913

36.Cui Y X, Biswal B K, Guo G, et al.Biological nitrogen removal from wastewater using sulphur-driven autotrophic denitrification.Applied microbiology and biotechnology, 2019, 103: 6023-6039.https://doi.org/10.1007/s00253-019-09935-4

37.Fan C, Zhou W, He S, et al.Sulfur transformation in sulfur autotrophic denitrification using thiosulfate as electron donor.Environmental Pollution, 2021, 268: 115708.https://doi.org/10.1016/j.envpol.2020.115708

38.Vázquez-Fernández A, Suárez-Ojeda M E, Carrera J. Review about bioproduction of Volatile Fatty Acids from wastes and wastewaters: Influence of operating conditions and organic composition of the substrate. Journal of Environmental Chemical Engineering, 2022, 10(3): 107917. https://doi.org/10.1016/j.jece.2022.107917

39.Wang D, Li Y, Cope H A, et al. Intracellular polyphosphate length characterization in polyphosphate accumulating microorganisms (PAOs): Implications in PAO phenotypic diversity and enhanced biological phosphorus removal performance. Water Research, 2021, 206: 117726. https://doi.org/10.1016/j.watres.2021.117726

40.Tayà C, Guerrero J, Suárez-Ojeda M E, et al. Assessment of crude glycerol for Enhanced Biological Phosphorus Removal: Stability and role of long chain fatty acids. Chemosphere, 2015, 141: 50-56.https://doi.org/10.1016/j.chemosphere.2015.05.067

41.Brahmacharimayum B, Mohanty M P, Ghosh P K.Theoretical and practical aspects of biological sulfate reduction: a review.Global NEST Journal, 2019, 21(2): 222-244.https://doi.org/10.30955/gnj.002577

42.Saad S A, Welles L, Lopez-Vazquez C M, et al.Sulfide effects on the anaerobic kinetics of phosphorus-accumulating organisms.of the World Congress on Anaerobic Digestion, Santiago de Compostela, Spain.2013.

43.Ikumi D S, Ekama G A.Plantwide modelling–anaerobic digestion of waste sludge from parent nutrient (N and P) removal systems.Water SA, 2019, 45(3): 305-316.

https://doi.org/10.17159/wsa/2019.v45.i3.6698

44.Lu X, Duan H, Oehmen A, et al.Achieving combined biological short-cut nitrogen and phosphorus removal in a one sludge system with side-stream sludge treatment.Water Research, 2021, 203: 117563. https://doi.org/10.1016/j.watres.2021.117563

45.Dahl C.A biochemical view on the biological sulfur cycle.Environmental Technologies to Treat Sulfur Pollution: Principles and Engineering, 2020, 2: 55-96.

https://doi.org/10.2166/9781789060966_0055

46.Gunarathne V, Rajapaksha A U, Vithanage M, et al.Hydrometallurgical processes for heavy metals recovery from industrial sludges.Critical Reviews in Environmental Science and Technology, 2022, 52(6): 1022-1062. https://doi.org/10.1080/10643389.2020.1847949

47.Chatla A, Almanassra I W, Abushawish A, et al.Sulphate removal from aqueous solutions: State-of-the-art technologies and future research trends.Desalination, 2023: 116615.https://doi.org/10.1016/j.desal.2023.116615

48.Plugge C M, Zhang W, Scholten J C M, et al. Metabolic flexibility of sulfate-reducing bacteria. Frontiers in microbiology, 2011, 2: 81.https://doi.org/10.3389/fmicb.2011.00081

49.Aüllo T, Ranchou-Peyruse A, Ollivier B, et al.Desulfotomaculum spp.and related gram-positive sulfate-reducing bacteria in deep subsurface environments.Frontiers in microbiology, 2013, 4: 362.https://doi.org/10.3389/fmicb.2013.00362

50.Qiu Y Y, Guo J H, Zhang L, et al.A high-rate sulfidogenic process based on elemental sulfur reduction: Cost-effectiveness evaluation and microbial community analysis.Biochemical engineering journal, 2017, 128: 26-32. https://doi.org/10.1016/j.bej.2017.09.001

51.Zhong W H, Zhu B T, Zhao C G, et al.Factors affecting nitrogen removal from aquaculture wastewater by Rhodob

---

## [Decision Letter · Decision Letter 1]

27 Aug 2024

Study on the Enhancement of Low Carbon-to-Nitrogen Ratio Urban Wastewater Pollutant Removal Efficiency by Adding Sulfur Electron Acceptors

PONE-D-24-01884R1

Dear Dr. Wei,

We’re pleased to inform you that your manuscript has been judged scientifically suitable for publication and will be formally accepted for publication once it meets all outstanding technical requirements.

Kind regards,

Guanglei Qiu

Academic Editor

PLOS ONE

Additional Editor Comments (optional):

Reviewers' comments:

Reviewer's Responses to Questions

**Comments to the Author**

1. If the authors have adequately addressed your comments raised in a previous round of review and you feel that this manuscript is now acceptable for publication, you may indicate that here to bypass the “Comments to the Author” section, enter your conflict of interest statement in the “Confidential to Editor” section, and submit your "Accept" recommendation.

Reviewer #2: All comments have been addressed

2. Is the manuscript technically sound, and do the data support the conclusions?

Reviewer #2: Yes

3. Has the statistical analysis been performed appropriately and rigorously? 

Reviewer #2: Yes

4. Have the authors made all data underlying the findings in their manuscript fully available?

Reviewer #2: Yes

5. Is the manuscript presented in an intelligible fashion and written in standard English?

Reviewer #2: Yes

6. Review Comments to the Author

Reviewer #2: The authors have revised the manuscript according to the reviewers' comments and suggestions. However, the figures were still blur. Please provide sharp figures before final publication.

7. PLOS authors have the option to publish the peer review history of their article (what does this mean?). If published, this will include your full peer review and any attached files.

Reviewer #2: No

---

## [Editor Report · Acceptance letter]

14 Oct 2024

PONE-D-24-01884R1 

PLOS ONE

Dear Dr. Wei, 

I'm pleased to inform you that your manuscript has been deemed suitable for publication in PLOS ONE. Congratulations! Your manuscript is now being handed over to our production team.

Kind regards, 

on behalf of

Dr. Guanglei Qiu 

Academic Editor

PLOS ONE